# Pharmacokinetics of Hydrogen During Hydrogen-Saturated Saline Infusion in Pigs

**DOI:** 10.3390/biomedicines13010234

**Published:** 2025-01-19

**Authors:** Masaki Shibuya, Masafumi Fujinaka, Mako Yonezawa, Natsumi Nishimura, Hitoshi Uchinoumi, Hiroshi Sunahara, Kenji Tani, Eiji Kobayashi, Motoaki Sano

**Affiliations:** 1Department of Medicine and Clinical Science, Graduate School of Medicine, Yamaguchi University, Ube 755-8505, Japan; mfuji@yamaguchi-u.ac.jp (M.F.); myone@yamaguchi-u.ac.jp (M.Y.); huchi@yamaguchi-u.ac.jp (H.U.); msano@yamaguchi-u.ac.jp (M.S.); 2Laboratory of Veterinary Surgery, Joint Faculty of Veterinary Medicine, Yamaguchi University, Yamaguchi 753-8511, Japan; sunahara@yamaguchi-u.ac.jp (H.S.); ktani@yamaguchi-u.ac.jp (K.T.); 3Department of Kidney Regenerative Medicine, The Jikei University School of Medicine, Tokyo 105-8461, Japan

**Keywords:** hydrogen gas, intravenous infusion, blood concentration, pig

## Abstract

**Background**: Hydrogen gas (H_2_) has been shown to be effective in the treatment of various clinical conditions, from acute illnesses to chronic illnesses. However, its clinical indications and the corresponding appropriate hydrogen delivery methods have yet to be determined. This is due to the fact that the pharmacokinetics and pharmacodynamics of hydrogen in each delivery method have not been experimentally proven. Here, we verified the pharmacokinetics of hydrogen after the infusion of hydrogen-saturated saline. **Methods**: Hydrogen-saturated saline was prepared and checked for sterility and component specifications. Hydrogen-saturated saline was administered intravenously (125 mL/h) through the left internal jugular vein of pigs, and the blood hydrogen concentration was measured over time. **Results**: It was confirmed that hydrogen can be safely mixed under pressure into intravenous solutions (pharmaceutical products) without the contamination of foreign substances by using a needle-less vial access cannula. No change in the PH or composition of the solution was observed due to hydrogen filling. The hydrogen concentrations of blood samples collected from the left internal jugular vein 3 cm to the heart from the tip of the infusion line were 6.4 (30 min), 4.7 (60 min), 4.9 (90 min), and 5.3 (120 min) ppb *w*/*w*, respectively. The hydrogen concentrations of blood samples collected from the right atrium were 0.7 (30 min), 0.5 (60 min), 0.7 (90 min), and 0.7 (120 min) ppb, respectively. The hydrogen concentration of blood samples collected from the right internal carotid artery were 0.1 (pre), 0.2 (30 min), 0.3 (60 min), 0.0 (90 min), and 0.0 (120 min) ppb *w*/*w*, respectively. **Conclusions**: We confirmed that hydrogen could be safely pressurized and filled into intravenous (pharmaceutical) solution without contamination by foreign objects using a needle-free vial access cannula. When saturated hydrogen saline was dripped intravenously, almost all of the hydrogen was expelled during its passage through the lungs and could not be supplied to the arterial side.

## 1. Introduction

In clinical trials, hydrogen gas (H_2_) has been shown to be safe [1] and effective in the treatment of various clinical conditions, from acute illnesses, such as myocardial infarction [2], post-cardiac arrest syndrome [3,4,5], subarachnoid hemorrhage [6], and COVID-19 pneumonia [7,8,9], to chronic illnesses, such as end-stage renal failure [10], chronic obstructive pulmonary disease [11,12], metabolic dysfunction-associated steatotic liver disease [13,14,15], cancer [16,17,18], cognitive impairment [19,20], Parkinson’s disease [21], and allergic and autoimmune diseases [22,23]. A wide variety of methods for delivering hydrogen have been tried, including inhalation, drinking hydrogen water, infusing hydrogen solution, bathing in hydrogen water, and electrolyzed water dialysis. However, the clinical indications and the corresponding appropriate hydrogen delivery methods have yet to be determined [24]. This is due to the fact that the pharmacokinetics and pharmacodynamics of hydrogen in each delivery method have not been experimentally proven. Clinical trials have shown that the intravenous administration of hydrogen solution may be effective in the treatment of acute brain damage due to cerebral infarction [25] and subarachnoid hemorrhage [6], as well as chronic rheumatoid arthritis [22] and allergic rhinitis [23]. However, there are no reports verifying the medical safety of the act of filling hydrogen into intravenous solutions (pharmaceuticals). In addition, there is no information on the pharmacokinetics of hydrogen itself when a hydrogen-containing solution is administered by drip infusion. In this study, we verified the required elements for filling hydrogen into an infusion solution—the sterility and stability of the solution components—in accordance with reliability compliance. Then, we verified the pharmacokinetics of hydrogen after the infusion of a hydrogen-saturated saline.

## 2. Materials and Methods

### 2.1. Preparation of Hydrogen-Saturated Saline

A hydrogen gas generator (H2JI1: Doctor’s Man Co., Ltd., Yokohama, Japan) was used as the hydrogen supply source. A hydrogen filling attachment was connected to the hydrogen gas generator, and the solenoid valve was opened to ensure that the inside of the piping contained only hydrogen. First, an 18G injection needle (Terumo, Tokyo, Japan) was inserted into the rubber stopper of a 500 mL Japanese Pharmacopeia saline solution soft bag (Otsuka Pharmaceutical Factory, Naruto, Japan, Lot. M4A85), and the bag was pressurized by hand to degas all air in the bag. A PTFE filter (S28PTB022DS: Microlab Scientific, Yueqing, Zhejiang, China) and filter attachment (Doctor’s Man) were connected one by one to the 18G needle (Figure 1A). The second hydrogen-saturated saline was prepared with the same method using a piercing plastic cannula (Safe Access™: CardinalHealth, Dublin, OH, USA) instead of an 18G needle (Figure 1B). The degassed saline bag, connected to the 18G injection needle, filter attachment, and PTFE filter, was set to fit snugly inside the pressure-resistant container, to prepare for filling the bag with hydrogen. Finally, the filter attachment was inserted into the hydrogen filling attachment connected to the hydrogen gas generator, and hydrogen gas was pressure-filled into the saline bag at 69 kPa (Figure 1C). After the hydrogen filling was completed, the 18G needle was removed from the saline bag and the bag was shaken for 30 s to prepare the hydrogen-saturated saline.

### 2.2. Confirmation of Sterility and Ingredient Specifications of Hydrogen-Saturated Saline

Bacterial culture tests and an analysis of solution components were performed on hydrogen-saturated saline. Using the methods described in the Japanese Pharmacopeia, the following categories were measured by the Japan Food Analysis Center (Shibuya-ku, Tokyo, Japan). The categories measured were sodium salt, chloride, insoluble particulates, pH, heavy metals, arsenic, endotoxin, insoluble foreign matter, sterility, ingredient content, color properties, general bacterial count (viable bacterial count), and coliform group.

### 2.3. Animal Experiments

Prior to this experiment, we used Fuji micromini pigs to verify whether it would be possible to insert catheters and collect blood samples and verify the blood hydrogen concentration in accordance with previous reports [26]. Medetomidine (40 μg/kg body weight) and midazolam (0.2 mg/kg body weight) were administered intramuscularly as a pretreatment to a female HI-COOP pig (8 weeks old; 50 kg). After securing an intravenous line, propofol (7 mg/kg body weight) was administered for anesthesia. For maintenance anesthesia, isoflurane was administered by inhalation under intubation using a vaporizer (KIV-7: Kimura Medical Instruments). The neck coat was shaved, and the skin surface was washed with warm water and disinfected with rubbing alcohol and veterinary povidone iodine solution. A longitudinal incision was made in the right neck to expose the right internal carotid artery and vein. A 16G catheter (1116-27PE; Covidien, Tokyo, Japan) was inserted through the right internal jugular vein and placed in the right atrium. A catheter was also inserted into the right internal carotid artery. A longitudinal incision was made in the left neck to expose the left internal jugular vein. A catheter was inserted into the left internal jugular vein as an intravenous drip line of hydrogen-saturated saline, and another catheter was inserted for blood sampling—the tip of which was fixed 3 cm to the heart from the tip of the drip line of hydrogen-saturated saline (Figure 2). A three-way stopcock was attached to each catheter, and the catheter was filled with normal saline containing heparin (Nipro, Lot. 24F01, Osaka, Japan). The surgical wounds were sutured and closed. Anesthesia, respiration, circulation, and body temperature were monitored in real time during the operation.

### 2.4. Measurement of Hydrogen Gas Concentration

The hydrogen gas concentration was measured by gas chromatography (TRIlyzer Biogas Analyzer mBA-3000; Taiyou Corporation, Joto-ku, Osaka, Japan), which was outsourced to Taiyou Corporation. One catheter (intravenous drip line), which was used as an intravenous administration line for hydrogen-saturated saline, was placed. Three catheters were also placed to simultaneously collect blood samples for hydrogen gas concentration measurement from three locations—a catheter placed 3 cm from the tip of the infusion line to the heart (left internal jugular vein), a catheter placed in the right atrium (right internal atrium), and a catheter placed in the right internal carotid artery (right internal carotid artery). One mL of blood was collected from the tip of the three catheters to measure the hydrogen concentration in the left internal jugular vein, right internal atrium, and right internal carotid artery. To ensure that the blood samples were collected at the same time, one experimentalist was assigned to each catheter to collect the samples. A 250 μL Hamilton syringe (Hamilton GS-4015-41525 Gas Tight Syringe 1700 Series 1725LTN 250 μL; Hamilton Company, Reno, NV, USA) was inserted into the rubber stopper of the three-way stopcock connected to the infusion line tubing of the hydrogen-saturated saline, and 100 μL of the hydrogen-saturated saline was collected before the infusion started. The hydrogen concentration (immediately after preparation) was measured. Before the intravenous administration of the hydrogen-saturated saline, 1 mL of blood was collected from the left internal jugular vein, right internal antrum, and right internal carotid artery, respectively, and the hydrogen concentration (pre) was measured.

Then, hydrogen-saturated saline was administered intravenously at a rate of 150 mL/h (3 mL/kg body weight/h) using an infusion pump (TOP-230V: TOP), and 1 mL of blood was collected from the left internal jugular vein, right internal atrium and right internal carotid artery at 30, 60, 90, and 120 min after administration started, respectively, and the hydrogen concentration (30 min, 60 min, 90 min, and 120 min) was measured. After the intravenous administration of hydrogen-saturated saline for 120 min, the infusion was terminated, a 250 μL Hamilton syringe was inserted into the rubber stopper of the three-way stopcock connected to the infusion line tubing of hydrogen-saturated saline, 100 μL of hydrogen-saturated saline was collected, and the hydrogen concentration after the administration was completed was measured.

## 3. Results

### 3.1. Sterility and Component Specifications of the First Hydrogen-Saturated Saline

The number of general bacteria was less than 100 cells/g. The coliform group was negative and no microorganisms were observed in the aseptic test. The analysis of the hydrogen-saturated saline showed that the color of the solution was clear and colorless, the pH was 5.6, and the sodium chloride content was 0.90 *w*/*v*%. Sodium, chloride, heavy metals, and arsenic were within the reference range, and endotoxin was less than 0.50 EU/mL. No insoluble particulates were observed, but easily detectable insoluble foreign materials were detected (Table 1).

### 3.2. Sterility and Component Specifications of the Second Hydrogen-Saturated Saline

The number of general bacteria was less than 100 cells/g. The coliform group was negative and no microorganisms were observed in the aseptic test. The analysis of the hydrogen-saturated saline showed that the color of the solution was clear and colorless, the pH was 5.8, and the sodium chloride content was 0.90 *w*/*v*%. Sodium and chloride were within the reference range, and endotoxin was less than 0.50 EU/mL. No insoluble foreign materials and particulates were detected (Table 2).

### 3.3. Contamination of Easily Detectable Insoluble Foreign Materials (Coring)

Easily detectable insoluble foreign materials were detected in the first hydrogen-saturated saline. They were considered to be rubber particles cored from the rubber stopper of the saline bottle by an 18G needle. Therefore, the second hydrogen-saturated saline was prepared with the same method using a piercing plastic cannula (Safe Access™: CardinalHealth) instead of an 18G needle (Figure 1B). No insoluble foreign materials were detected in the second hydrogen-saturated saline.

### 3.4. Blood Pressure, Heart Rate, and Percutaneous Oxygen Saturation During Hydrogen-Saturated Saline Infusion

Blood pressure, heart rate, and oxygen saturation were monitored at 30 min intervals for 120 min during hydrogen-saturated saline infusion in the pig model. Systolic blood pressure at prior infusion (pre), 30 min (30 min), 60 min (60 min), 90 min (90 min), and 120 min (120 min) was 101, 99, 101, 100, and 94 mmHg, respectively. Diastolic blood pressure at pre, 30 min, 60 min, 90 min, and 120 min was 44, 44, 42, 43, and 40 mmHg, respectively. Mean blood pressure at pre, 30 min, 60 min, 90 min, and 120 min was 60, 59, 58, 58, and 54 mmHg, respectively. Heart rate at pre, 30 min, 60 min, 90 min, and 120 min was 60, 59, 58, 58, and 54 beats per minute, respectively. Percutaneous oxygen saturation at pre, 30 min, 60 min, 90 min, and 120 min was 99, 98, 98, 97, and 97%, respectively (Table 3, Figure 3).

### 3.5. Hydrogen Gas Concentration

Before intravenous administration, the hydrogen concentration of the hydrogen-saturated saline collected from the infusion line tube was 1646.6 ppb *w*/*w*. The hydrogen concentration of blood samples collected from the left internal jugular vein, right atrium, and right internal carotid artery prior to intravenous administration of the hydrogen-saturated saline was 0.5, 0.1, and 0.1 ppb *w*/*w*, respectively (pre). Hydrogen-rich saline was administered intravenously at a rate of 150 mL/h (3 mL/kg body weight/h). The hydrogen concentration of blood at 30 min was 6.4, 0.7, and 0.2 ppb *w*/*w*, respectively. The hydrogen concentration of blood at 60 min was 4.7, 0.5, and 0.3 ppb *w*/*w*, respectively. The hydrogen concentration of blood at 90 min was 4.9, 0.7, and 0.0 ppb *w*/*w*, respectively. After 120 min, the hydrogen concentration of blood was 5.3, 0.7, and 0.0 ppb *w*/*w*, respectively. Soon after finishing intravenous administration, the hydrogen concentration of the hydrogen-saturated saline collected from the infusion line tube was 933.0 ppb *w*/*w* (Table 4, Figure 4).

## 4. Discussion

### 4.1. The Medical Safety of Filling Hydrogen to Intravenous Drip Solution (Pharmaceuticals)

Hydrogen has a low solubility and a high diffusivity [28,29,30]. If left at room temperature and atmospheric pressure, hydrogen will diffuse from the drip bag to the air. In the experiment, the hydrogen concentration, which was 1646.6 ppb when the drip started, had decreased to 933.0 ppb after two hours. Therefore, it is preferable to fill the solution with hydrogen just before starting an intravenous drip. We previously developed a new device that can safely transport hydrogen gas using a hydrogen storage alloy canister, which can instantaneously pressurize a high concentration of hydrogen gas into a container of organ storage solution [31]. Using this technology, we have demonstrated that the kidneys of elderly mini-pigs, which have suffered severe injury due to circulatory arrest, can maintain their urinary excretion function after being transplanted into another elderly mini-pig by exposing them to hydrogen gas in organ storage solution [32].

### 4.2. Safety When Injecting Hydrogen Gas Through a Rubber Stopper

Coring is the retention of a material from a medication vial into the needle and syringe, which can ultimately be transfused into a patient, causing adverse outcomes. It occurs when a needle shears out cores from a rubber closure as it punctures the closure. The prevention of coring has largely focused on encouraging visual inspection, changing the angle of entry into the rubber stopper, orienting the drawing needle bevel upward, controlling the drawing pressure, or changing the drawing needle and syringe size [33].

We use Safe Access™, a needle-less vial access side hole cannula, which is specifically designed to access rubber stopper medication vials without the risk of needlestick injury. It also eliminates coring and particulate by piercing—not puncturing—the vial.

### 4.3. PK of Hydrogen After Hydrogen-Saturated Saline Infusion

Two catheters were inserted into the left internal jugular vein. One catheter was placed as an infusion line for hydrogen-saturated saline, and the other catheter was placed 3 cm to the heart from the tip of the infusion line for blood sampling. The catheter for blood collection was inserted from the right internal jugular vein and was indwelling in the right atrium. A catheter was also placed in the right internal carotid artery. The hydrogen concentration of the hydrogen-saturated saline taken from the infusion line before the start of the infusion was 1646.6 ppb. Since the saturated concentration of hydrogen at normal temperature and pressure is approximately 1.6 ppm (1600 ppb), this means that the hydrogen-containing saline was saturated. The hydrogen concentration in the left internal jugular vein 3 cm from the tip of the infusion line was 6.4 ppb at 30 min, 4.7 ppb at 60 min, 4.9 ppb at 90 min, and 5.3 ppb at 120 min, respectively. This means that the hydrogen concentration in the left internal jugular vein was diluted by the blood by about 250 to 300 times. The hydrogen concentration in the right atrium was 0.7 ppb at 30 min, 0.5 ppb at 60 min, 0.7 ppb at 90 min, and 0.7 ppb at 120 min. This means that the hydrogen concentration in the right atrium was diluted by the blood by about 2300 times. Almost all of the hydrogen in the blood was expelled into the exhaled air during passage through the lungs and could not be delivered into the right carotid artery. The same 500 mL dose of saturated glucose solution, injected in 2 min into the upper small intestine via a gastrointestinal tube, was able to maintain the hydrogen concentration in the portal vein to some extent for more than 2 h [34]. This may be because hydrogen is gradually absorbed through the intestinal tract, and the absorbed hydrogen is concentrated in the portal vein via the jejunal vein. On the other hand, when hydrogen-saturated saline is administered via intravenous injection, it is rapidly diluted by the blood, and hydrogen is progressively withdrawn from the blood vessels, mainly through the lungs. Even near the top of the infusion line, the blood hydrogen concentration increased only to 0.3–0.4%. It can be concluded that in order to deliver hydrogen throughout the body via intravenous infusion, a large amount of hydrogen solution must be continuously infused at a fast infusion rate. In particular, a situation can be envisioned in which hydrogen is filled in the replacement fluid of hemodiafiltration. In the field of hemodialysis, a hemodialysis system that provides a stable supply of hydrogen-dissolved dialysis water using water electrolysis technology (electrolyzed water dialysis) has been developed, and the clinical usefulness of this system has been reported in clinical studies in the short to medium term [35,36].

## 5. Conclusions

It was confirmed that hydrogen could be safely pressurized into an intravenous drip solution (pharmaceuticals) without contamination by foreign substances using a needle-less vial access cannula. No change in the PH or composition of the solution was observed by filling hydrogen. Hydrogen infusion may not be an adequate delivery method for hydrogen in a normal peripheral intravenous infusion. It was suggested that if both the dosage and the speed of administration could be scaled up, it could be a practical method of hydrogen delivery.

## 6. Patents

M. Sano is the registered inventor of the following patents jointly filed by Keio University and Taiyo Nippon Sanso: hydrogen mixed gas supply device for medical purposes (patent number: 5631524); medicinal composition for improving prognosis after restart of patient’s own heartbeat, and medicinal composition for improving and/or stabilizing circulatory dynamics after the onset of hemorrhagic shock; and three other patents (whose names are translated into English)—pharmaceutical compositions for reducing weight loss after organ harvesting (joint application with Keio University and Taiyo Nippon Sanso); method for generating organ preservation solution containing hydrogen and organ preservation solution containing hydrogen (joint application with Keio University and Doctors Man; application number PCT/JP2019/045790).

## Figures and Tables

**Figure 1 biomedicines-13-00234-f001:**
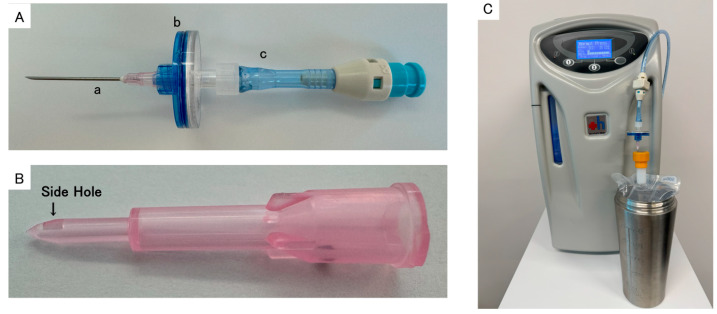
(**A**): a. An 18G injection needle (NN-2138R; Terumo); b. PTFE filter (S28PTB022DS; Microlab Scientific); c. filter attachment (Doctor’s Mann). (**B**): Piercing side hole plastic cannula (Safe Access™: CardinalHealth). (**C**): The degassed saline bag, connected to the 18G injection needle, filter attachment, and PTFE filter, was set to fit snugly inside the pressure-resistant container.

**Figure 2 biomedicines-13-00234-f002:**
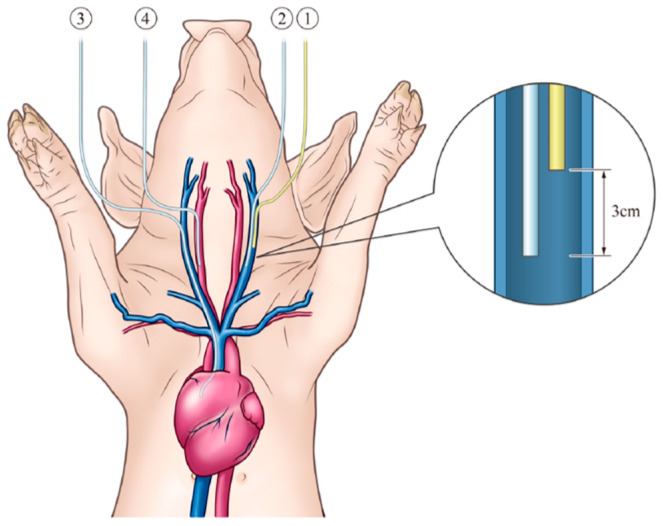
① One catheter was inserted into the left internal jugular vein as an intravenous drip line of hydrogen-saturated saline. ② Another catheter was inserted for blood sampling—the tip of which was fixed 3 cm to the heart from the tip of the drip line of hydrogen-saturated saline. ③ The third catheter was inserted into the right internal carotid artery. ④ The fourth catheter was inserted through the right internal jugular vein and placed in the right atrium.

**Figure 3 biomedicines-13-00234-f003:**
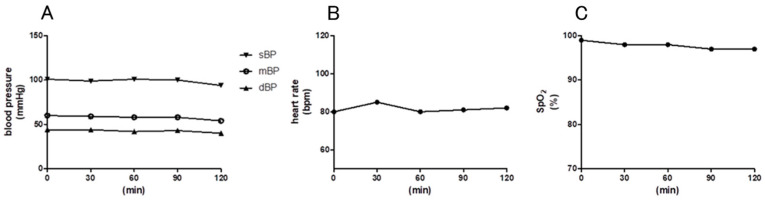
(**A**): Blood pressure, (**B**): heart rate, and (**C**): SpO_2_ during hydrogen-saturated saline infusion.

**Figure 4 biomedicines-13-00234-f004:**
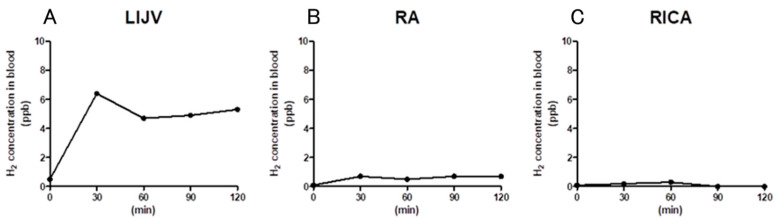
Time course of H2 blood concentration during the intravenous infusion of hydrogen-saturated saline. (**A**): Left internal jugular vein, (**B**): right atrium, and (**C**): right internal carotid artery.

**Table 1 biomedicines-13-00234-t001:** Sterility and component specifications of first hydrogen-saturated saline.

Analysis Item: Saline Solution *		Result
Confirmation test	sodium salt (1)	reference range
	sodium salt (2)	reference range
	chloride (1)	reference range
	chloride (2)	reference range
pH		5.6
Endotoxin		less than 0.50 EU/mL without response interference factor
Insoluble extraneous material **		easily detectable
Insoluble particles **	more than 10 μm	0/mL
	more than 25 μm	0/mL
Sterility **	liquid thioglycolic acid medium	no microorganisms
	soybean–casein digest medium	no microorganisms
	SCDLP agar plate medium	bacteria: less than 100/g
	LB medium	coliform bacteria: negative/1 g
Content	sodium chloride	0.90 *w*/*v*%
Characteristics	color/shape	clear and transparent liquid

*. The 18th edition of the Japanese Pharmacopeia (before partial revision by Notification No. 355 of the Ministry of Health, Labour and Welfare in 2022) [27]. **. One specimen tube was tested, as chosen by the client.

**Table 2 biomedicines-13-00234-t002:** Sterility and component specifications of second hydrogen-rich saline.

Analysis Item: Saline Solution *		Result
Confirmation test	sodium salt (1)	reference range
	sodium salt (2)	reference range
	chloride (1)	reference range
	chloride (2)	reference range
pH		5.8
Endotoxin		less than 0.50 EU/mL
Insoluble extraneous material **		no insoluble extraneous material
Insoluble particles **	more than 10 μm	0/mL
	more than 25 μm	0/mL
Sterility **	liquid thioglycolic acid medium	no microorganisms
	soybean–casein digest medium	no microorganisms
	SCDLP agar plate medium	bacteria: less than 100/g
	LB medium	coliform bacteria: negative/1 g
Content	sodium chloride	0.90 *w*/*v*%
Characteristics	color/shape	clear and transparent liquid

*. The 18th edition of the Japanese Pharmacopeia (before partial revision by Notification No. 355 of the Ministry of Health, Labour and Welfare in 2022) [27], **. One specimen tube was tested, as chosen by the client.

**Table 3 biomedicines-13-00234-t003:** Blood pressure, heart rate, and SpO_2_ during hydrogen-saturated saline infusion.

Parameter	Pre	Time After Initiation of Infusion (min)			
		30	60	90	120
sBP (mmHg)	101	99	101	100	94
dBP (mmHg)	44	44	42	43	40
mBP (mmHg)	60	59	58	58	54
HR (bpm)	80	85	80	81	82
SpO_2_ (%)	99	98	98	97	97

sBP: systolic blood pressure; dBP: diastolic blood pressure; mBP: mean blood pressure; HR: heart rate; SpO_2_: percutaneous oxygen saturation.

**Table 4 biomedicines-13-00234-t004:** Time course of H2 blood concentration (ppb *w*/*w*) during the intravenous infusion of hydrogen-saturated saline.

	Pre	Time After Initiation of Infusion (min)			
		30	60	90	120
LIJV	0.5	6.4	4.7	4.9	5.3
RA	0.1	0.7	0.5	0.7	0.7
RICA	0.1	0.2	0.3	0.0	0.0

## Data Availability

Data is contained within the article.

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
