# Peer review of "Pharmacokinetics of Hydrogen During Hydrogen-Saturated Saline Infusion in Pigs"

_biomedicines, 2025, doi:10.3390/biomedicines13010234_

Round 1
Reviewer 1 Report
Comments and Suggestions for Authors
In the manuscript entitled “The blood concentration of hydrogen gas and safety of hydrogen-saturated saline infusion therapy,” the authors verified the required elements for filling hydrogen into an infusion solution: sterility and stability of the solution components due to the inclusion, in accordance with the reliability compliance, and verified the pharmacokinetics of hydrogen after infusing of hydrogen-saturated saline. It's an exciting work due to the potential of hydrogen in treating several conditions. In this way, to improve the quality of the manuscript, the authors must pay attention to the following aspects:
- In the line 35, describe COPD and MASLD
- The sentence “This is due to the fact that there is no Pharmacodynamics based on the Pharmacokinetics of hydrogen itself” does not make sense. Please improve it.
- The Tables appear to have been taken from another publication. Please write it in the paper.
Reviewer 2 Report
Comments and Suggestions for Authors
The research design seems appropriate for the stated aim of verifying the pharmacokinetics of hydrogen after infusion of hydrogen-saturated saline. It clearly identifies a gap in the existing literature: the lack of pharmacokinetics and pharmacodynamics data for hydrogen administration via intravenous infusion. The decision to study the sterility, stability, and pharmacokinetics of hydrogen-saturated saline aligns well with the stated gaps and clinical potential of hydrogen therapy.
The results for hydrogen concentrations over time are presented clearly with numerical data, which gives a good understanding of the findings. The conclusion that hydrogen infusion might not be adequate for peripheral intravenous delivery is supported by the observed low and declining hydrogen concentrations. The suggestion about scaling up dosage and administration speed is logical but speculative, as no data in the abstract directly supports these modifications. The confirmation of safety (no contamination or changes in pH) is supported by the results provided.
The discussion highlights how the study design aligns with its objectives, particularly focusing on the challenges of hydrogen stability, infusion methods, and pharmacokinetics (PK). The use of multiple catheter placements to monitor hydrogen concentrations at different points in the circulatory system is appropriate for tracking hydrogen diffusion and dilution. The discussion provides detailed explanations of hydrogen concentration measurements and the infusion setup, particularly the catheter placements. The use of needleless vial access technology (Safe Access™) to mitigate risks such as coring and contamination is described clearly.
The hydrogen concentrations at various anatomical sites and time points are clearly quantified, which aids in understanding the rapid dilution and exhalation of hydrogen. The loss of hydrogen over time (e.g., from the drip bag) is reported in practical terms, offering actionable insights (e.g., hydrogen should be infused promptly after preparation). The conclusion that hydrogen is rapidly diluted in the bloodstream and expelled through the lungs is directly supported by the data. The suggestion to continuously infuse large amounts of hydrogen solution at a fast rate for effective delivery is a logical extension of the observed rapid dilution. The potential application of hydrogen infusion in hemodialysis systems is supported by references to related technologies and prior studies.
